# Why Have China's Poverty Eradication Policy Resulted in the Decline of Arable Land in Poverty-Stricken Areas?

**Rong Ran** [1], **Lei Hua** [1], **Tingrou Li** [1], **Yejing Chen** [1] **and Junfu Xiao** [2,*]

[1] School of Public Policy and Administration, Chongqing University, Shapingba District, Chongqing 400044, China; ranrong@cqu.edu.cn (R.R.); 202201021012@stu.cqu.edu.cn (L.H.); 202101021013@cqu.edu.cn (T.L.); 20190007@cqu.edu.cn (Y.C.)

[2] School of Economics and Business Administration, Chongqing University, Shapingba District, Chongqing 400044, China

[*] Correspondence: xiaojunfu@cqu.edu.cn

**Abstract:** Arable land resources are the basic livelihood security for people in poverty-stricken areas, and poor people are prone to uncontrolled expansion of arable land and exogenous ecological damage to secure their livelihoods. To avoid this vicious cycle, China's poverty eradication policy requires greater management and restoration of arable land in poverty-stricken areas, but it is unknown what impacts it may bring. Therefore, this study examines the impact of policy implementation on arable land by the Difference-in-Differences (DID) model and uses the mediating and moderating models to test the policy's mechanism on arable land. The results reveal that the policy significantly reduces the arable land, and the results remain robust, controlling for potential endogeneity variables and robustness tests. What's more, the results of the mediating and moderating effect models indicate that the reduction of arable land is mainly due to the increase of unit yield, guiding the local people to abandon degraded land and to carry out ecological restoration, thus reducing the arable land area and achieving sustainable development. Finally, recommendations are proposed from the perspective of human-land coordination.

**Keywords:** arable land; China's poverty eradication policy; poverty-stricken areas; difference-in-differences





## 1. Introduction

Land resources, one of the most important factors of natural resources, are an essential production and living element for people to survive and develop [1,2]. Among them, arable land, as the essence of land resources, is the basic element of agricultural production and the main source of food provision [3,4]. Therefore, the protection and utilization of arable land resources is of special significance to ensure food security, ecological safety, and sustainable use of resources [5–7]. Previous studies reveal that about a quarter of the national poverty-stricken counties in China are subject to unfavorable constraints on land resources, mainly in terms of poor topographical conditions, poor arable land resources, scarcity and scattering, and serious degradation, which is not conducive to large-scale centralized operation and seriously affect the efficiency of agricultural production, thus affecting the livelihoods and incomes of the poor [8–10]. The optimization and integration of arable land resources serve as an important platform and foundation for the socio-economic development of poverty-stricken areas and a basic livelihood guarantee for the population in poverty-stricken areas [11,12]. The efficient and sustainable use of arable land can effectively solve the food problem in poverty-stricken areas, promote regional agricultural development, and improve farmers' living standards, which is of great significance to alleviating and eliminating regional poverty [13].

In the previous phases of poverty eradication, people in poverty-stricken areas would pursue the quantity of arable land too much and adopt rough farming to earn their livings [14], which led to the overuse of arable land and constantly reduced the resource and

environmental carrying capacity of arable land in poverty-stricken areas, resulting in the deterioration of their ecological environment [15]. For example, the massive application of chemical fertilizers, pesticides, and mulch to promote food production has increased the degree of exogenous environmental stresses such as soil erosion, pollution, soil salinization and soil sanding [16,17]. After the rough cultivation mentioned above, the original arable land is faced with the depletion of arable land and a decline in unit yields [18], which in turn fails to satisfy the livelihoods of the poor. Therefore, they, in turn, will continue to opt for deforestation and clearing of new arable land [19,20], abandoning the former arable land, thus resulting in the uncontrolled expansion of the arable land. At the same time, the environmental problems facing the original arable land have not been dealt with appropriately [21]. It leads to the continuous destruction of the ecological environment and puts poor areas into a vicious circle [22].

China's poverty eradication policy requires that ecological protection be given top priority and that new methods of ecological poverty alleviation be explored alongside economic development to eliminate poverty, which requires that ecological damage be eliminated in agricultural development in poverty-stricken areas [23]. For instance, including the prohibition of new land reclamation for cultivation, the enhancement of complementary water resources and irrigation infrastructure, the dissemination of novel agricultural production technologies to increase crop yields per unit of arable land, the implementation of soil testing and tailored fertilizer application techniques to mitigate non-point source pollution and prevent soil compaction, ecological restoration of degraded cropland, and the reinforcement of arable land quality protection and enhancement [24]. However, only a ban on new arable land reclamation to curb the uncontrolled expansion of arable land will likely affect people's livelihood in poverty-stricken areas because their main income is from agricultural production [25–27]. Therefore, current land and poverty alleviation research focuses more on compensation for fallowing and economic effects brought by land rectification and mostly concentrates on theoretical analysis and case summaries [28]. There is a lack of research on the impact of China's poverty eradication policies on arable land in poverty-stricken areas, especially in poverty-stricken mountainous areas. Moreover, the mechanism by which China's poverty eradication policies affect arable land in poverty-stricken areas has not been elucidated.

Based on the above analysis, this study takes the poverty-stricken counties removed from poverty in the contiguous poverty area in Qinba Mountain in 2019 as the sample and the implementation of the poverty eradication policy in 2015 as the time point [1]. The DID model is used to assess the changes in the amount of arable land in the poverty-stricken counties of the contiguous impoverished areas in the Qinba Mountains during the process of eliminating extreme poverty, and the mediating and moderating effect models are used to test the mechanisms of policy effects on arable land. The specific objectives of the study are as follows: (1) to verify whether the implementation of the poverty eradication policy has an impact on the change of arable land in the contiguous impoverished areas in the Qinba Mountains by using the DID model; (2) using a mediated effects model to test what factors influence the area of arable land; (3) to test whether the change of arable land areas effectively improves the ecological environment quality by using the regulating effect model.

## 2. Theoretical Mechanisms

Since the natural environment base of poverty-stricken areas is relatively poor compared with other areas, the poor arable land, poor quality endowment conditions, and spatial distribution show fragmentation characteristics. Thus the land productivity is relatively low, which is extremely unfavorable to the agricultural development of poor people [29,30]. So, the improvement of farming yield depends largely on the input of agricultural production materials such as chemical fertilizers, pesticides, and mulch. For example, the excessive and irrational use of fertilizers, pesticides and mulch can cause agricultural surface pollution and exacerbate the problem of declining arable land quality,

thus creating a vicious cycle of increased reliance on fertilizers, pesticides and mulch for agricultural production inputs [31,32]. In addition, irrational irrigation methods, such as heavy irrigation the imbalance focus on the use of land rather than the maintenance of land are also important causes of increased arable land degradation [33]. With the rapid degradation of arable land quality, the population in poverty-stricken areas will further deforest and clear new arable land to make a living. These phenomena are undoubtedly a vicious circle of environmental destruction and ecological resource sacrifice for poverty-stricken areas, which will cause huge externalities to biodiversity and climate change and harm sustainable development [34,35].

Existing research has also delved into the unregulated expansion and unsustainable utilization of arable land in impoverished regions from a supply and demand perspective [36–38]. These research findings suggest that impoverished areas typically experience higher population growth rates. Agricultural households in these regions require a sufficient supply of food and income to meet the needs of their family members [39]. Consequently, this increases the demand for arable land to enhance crop yields and land utilization [40]. They may be deficient in modern agricultural technologies, efficient farming practices, and comprehensive infrastructure. This makes it more challenging for them to achieve high output and sustainable agricultural practices, rendering them unable to meet their subsistence requirements and exacerbating the demand for arable land [41,42]. However, the available arable land resources in impoverished regions are typically limited and beset with issues such as degradation, infertility, or fragmentation. As a result, farmers are compelled to rely on the finite land available for agricultural production, leading to overutilization and cultivation. Consequently, this results in a deterioration of soil quality, nutrient loss, and intensified soil erosion. This not only hinders the growth and yield of crops, resulting in unregulated expansion and unsustainable utilization of cropland but also engenders adverse effects on the ecological system's health and the sustainable livelihoods of farmers [43,44].

China's poverty eradication policy requires that ecological protection be given top priority and that new methods of ecological poverty alleviation be explored alongside economic development to eliminate poverty, which requires that poverty-stricken areas eliminate ecological damage in agricultural development. In the policy implementation process, local governments have undertaken various measures to alleviate or eliminate the prevailing conditions of cropland infertility, degradation, and unregulated expansion. For example, promote major projects to improve and restore arable land, including enhancing the renovation of low-and-medium-yielding fields or the construction of high-standard farmland according to local conditions, trying to increase the unit food production of arable land [45,46]. Advanced agricultural technologies such as efficient water-saving irrigation and rational fertilization have been introduced in arable land preparation and management. These can enhance crop yields on limited arable land and reduce the demand for new cropland expansion [47].

Additionally, the adoption of advanced agricultural machinery and equipment has improved labor productivity, reducing the reliance on manual cultivation [48,49]. Furthermore, with the widespread dissemination of agricultural technology, farmers can increasingly rely on diversified agricultural production to meet various demands. This can lead to improved economic returns, livelihood satisfaction, and reduced demand for cropland, significantly mitigating the likelihood of unregulated cropland expansion in impoverished areas. Consequently, following the increase in per-unit grain production, farmers may emphasize land conservation and sustainability. They may implement measures to reduce soil erosion, unregulated expansion, and unsustainable land utilization, ensuring the land can maintain high productivity over the long term [50,51]. This shift can obviate the need for large-scale land expansion, enabling better management and conservation of existing land resources to ensure long-term agricultural sustainability.

In addition to implementing the aforementioned high-standard farmland measures, impoverished regions adopt various strategies for the governance of degraded and deserti-

fied cropland [52–54]. These strategies include cropland retirement and the enhancement of windbreak and sand-fixation projects for desertified cropland, which facilitate ecological restoration and soil management. Degradation and desertification lead to a decline in land quality, severe water, and nutrient loss, rendering the land unsuitable for crop cultivation. Through afforestation, grassland restoration, and soil conservation, soil quality can be improved, enhancing moisture retention capacity and providing a more favorable growth environment for crops while concurrently restoring the ecological environment [55,56]. However, these measures may require allocating a portion of cropland for ecological restoration, reducing arable land area. For instance, the construction of terraces on sloping terrain, vegetation belts, and protective forest belts is undertaken to slow down water flow and prevent soil erosion. Rational irrigation management measures, such as drip irrigation and spray irrigation, can also be implemented to reduce water wastage and enhance irrigation efficiency [57,58]. Such actions serve the dual purpose of safeguarding and restoring the ecological environment while ensuring high-quality agricultural production.

Simultaneously, following the rehabilitation of degraded and desertified cropland, local governments promote adjustments to agricultural structures to reduce excessive land utilization. This may involve guiding farmers to diversify into other industries or altering crop planting patterns, such as promoting drought-resistant crops, crop rotation and fallow practices, and developing multi-story agroforestry [59]. These measures not only secure income for the impoverished population but also reduce the overexploitation of cropland, safeguarding the ecological environment from degradation and mitigating instances of unregulated cropland expansion due to livelihood constraints. Adopting a multi-faceted approach that combines ecological restoration, sustainable agricultural practices, and agricultural structural adjustments ultimately promotes both environmental sustainability and the well-being of their communities [60,61].

Therefore, this paper proposed the following hypotheses:

**H1.** *The poverty eradication policy will reduce the amount of arable land in poverty-stricken areas.*

**H2.** *The poverty eradication policy will mitigate the negative impact of reducing the amount of arable land in poverty-stricken areas by increasing the unit of food production.*

**H3.** *The poverty eradication policy will reduce the amount of arable land in poverty-stricken areas by improving the quality of the ecological environment.*

## 3. Materials and Methods

### 3.1. Study Area

The Qinba Mountain region is located at the junction of six provinces and regions in China (As shown in Figure 1), namely Henan, Sichuan, Chongqing, Hubei, Gansu, and Shaanxi. This contiguous area encompasses 75 nationally designated impoverished counties [62,63]. Simultaneously, the Qinba Mountain region holds pivotal significance as a designated key ecological functional area in China's ecological zoning framework, as well as a critical region for the ecological barrier of the upper Yangtze River basin. Owing to its harsh natural environment, intricate topography, and severe land degradation resulting from historical haphazard land utilization practices during agricultural development, the Qinba Mountain region confronts a series of formidable ecological challenges [64]. These challenges include a disorderly land use structure, ecological landscape deterioration, exacerbated non-point source pollution from agriculture, etc. Such issues can directly or indirectly obstruct wildlife migration corridors, leading to biodiversity loss, diminishing water resource conservation functions, and impeding the normal flow of ecosystem services in the Qinba Mountain region [65,66].

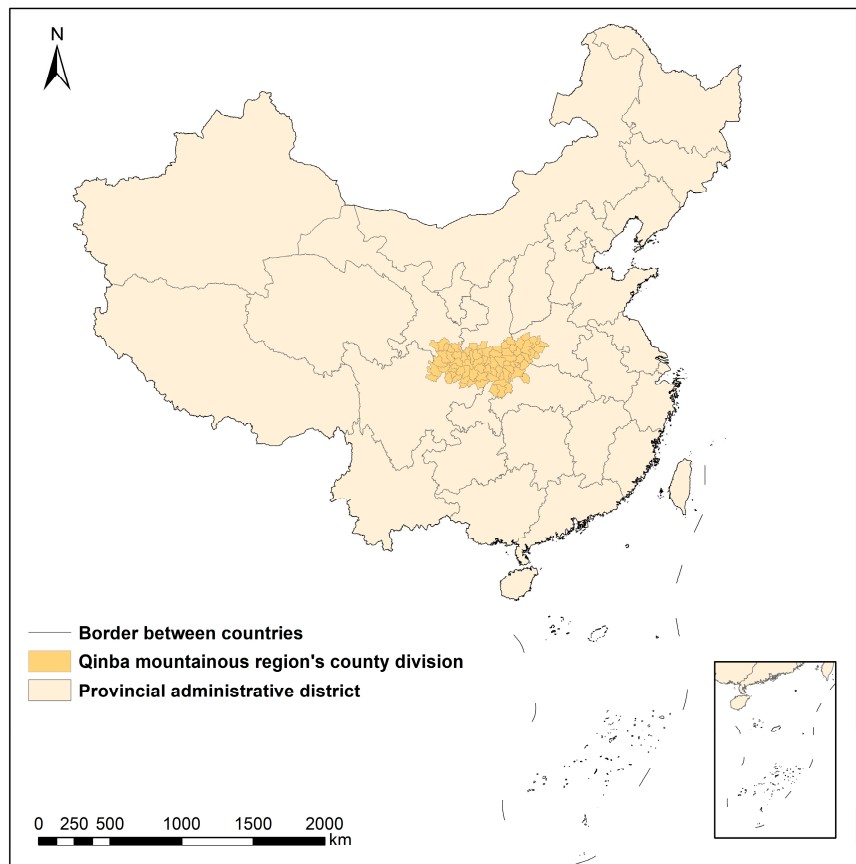

**Figure 1.** Qinba mountain area schematic diagram.

*3.2. Method*

The research question addressed in this paper pertains to the impact of poverty alleviation policies on the cultivated land area in contiguous impoverished regions of the Qinba Mountain area. Currently, most relevant studies assessing policy effects employ the DID model, a method designed to mitigate endogeneity issues as much as possible, thereby identifying the net effects of policies. In this study, we regard poverty alleviation policies as a quasi-natural experiment. By applying the DID model while controlling for relevant control variables, we compare changes in cultivated land area between impoverished counties and non-impoverished counties. This approach enables us to accurately discern causal relationships, remove temporal trend variations, and estimate the influence of policies on cultivated land area in impoverished regions, ensuring that changes in cultivated land area in impoverished counties are attributed to the policies. Based on the previous studies, the following models are obtained [67,68]:

$$\text{CLand}_{it} = \beta_0 + \beta_1 \text{Treat} \cdot \text{T} + \beta_2 \text{control}_{it} + \eta_t + \mu_i + \varepsilon_{it} \tag{1}$$

where $\text{CLand}_{it}$ represents the arable land area of the $t_{th}$ poverty-stricken county within the Qinba Mountain region in year i; Treat is used to distinguish the control group from the experimental group in the sample (the experimental group is poverty-stricken counties within the Qinba Mountain region and the control group is non-poverty-stricken counties within the Qinba Mountain region), T is used to distinguish before and after the implementation of the policy, and Treat·T is the core explanatory variable of this paper; if the policy occurs and the county is a poverty-stricken county that will be out of poverty in 2019, then Treat T = 1, otherwise 0; $\text{Control}_{it}$ represents the selected control variables, i.e., economic level, population density, fiscal revenue level, fiscal expenditure level, value added of primary industry, and precipitation; $\eta_t$ controls for time-level characteristics that do not vary with

region, such as changes in the macroeconomic situation; $\mu_i$ controls for individual-level characteristics that do not vary over time; $\varepsilon_{it}$ represents the random disturbance term; the coefficien $\beta_1$ represents the effect of the poverty eradication policy on poverty-stricken counties in the Qinba Mountains, which is the core coefficient in this paper.

### 3.3. Variables and Data Sources

Explanatory variable. The area of cultivated land in poverty-stricken areas, $CLand_{it}$, is the explanatory variable.

Core explanatory variables. The cross-multiplication term Treat·T is the core explanatory variable, representing whether the poverty-stricken counties in the Qinba Mountains implement the poverty eradication policy. Among them, Treat is the policy dummy variable. If the sample counties are poverty-stricken counties that will be out of poverty in 2019, then Treat = 1, otherwise 0; T is the experimental period dummy variable, if the time is after the implementation of the policy in 2015 (including 2015), then T = 1, otherwise 0. The coefficient of the cross-product term Treat·T, $\beta_1$, represents the net impact of the policy on the arable land area of poverty-stricken counties in the Qinba Mountains, and Treat·T is assigned to 1 only when the ith county is a national-level poverty-stricken county in the Qinba Mountains that escape poverty in 2019 and t ≥ 2015, otherwise 0.

Control variables. Arable land area is influenced by a variety of factors. Based on the previous studies, this paper selects the EL, PRCP, PD, FEL, FRL, and AVPI as the control variables (Table 1).

**Table 1.** Variables description.

| Variable | Variable Symbol | Description |
|---|---|---|
| The Cultivated land area | CLand | Obtaining the area of cultivated land in poor areas through arcGis. (Units: km$^2$) |
| Cross term | Treat·T | Representing whether the poverty-stricken counties in the Qinba Mountains implement the poverty eradication policy. |
| The level of economic development | EL | Reflecting the region's current state of economic development, measured by the natural logarithm of GDP [69,70]. (Units: 10,000 yuan) |
| Precipitation | PRCP | Area precipitation [71]. (Units: Millimeter) |
| Population density | PD | Measured as the ratio of the total population at the end of the year to the size of the administrative area [72,73]. (Units: 10,000 people/km$^2$) |
| Fiscal expenditure level | FEL | Measured as the natural logarithm of general budget expenditures of local finances [74]. (Units: 10,000 yuan) |
| Fiscal revenue level | FRL | Measured as the natural logarithm of general budget revenues of local finances [75]. (Units: 10,000 yuan) |
| The development of the primary sector | AVPI | Measured by the natural logarithm of the value added of the primary sector [76]. (Units: 10,000 yuan) |

This paper assesses the policy effects of poverty eradication policy by using panel data of 172 districts and counties (county-level cities) in contiguous impoverished areas in the Qinba Mountains from 2011 to 2019. Considering that poverty-stricken counties in the Qinba region have been removed from poverty one after another from 2016 to 2018, the sample excludes counties successfully removed from poverty in 2016–2018 to ensure that the empirical results are not affected. In this paper, the 46 poverty-stricken counties

that successfully escaped from poverty in 2019 are selected as the treatment group, and the sample of districts and counties (county-level cities) in the remaining sample is taken as the control sample, using the national implementation of poverty eradication policy in 2015 as the external policy shock point. Relevant socioeconomic data are obtained from the annual China County (City) Social and Economic Statistical Yearbook, China County Statistical Yearbook, and district and county statistical bulletins. The Qinba mountainous area county administrative boundary data from the National Earth System Science Data Center, National Science & Technology Infrastructure of China (http://www.geodata.cn, accessed on 22 September 2023). The LUCC data is obtained from the annual dataset of land use and cover (https://doi.org/10.5194/essd-2021-7, accessed on 1 July 2020) from 1985 to 2020, hosted by Wuhan University and based on the Landsat images. The spatial resolution of these data is 30 m, and the spatial resolution is 30 m. The accuracy of the data is verified to be 79% by the visually interpreted independent samples and the third-party test samples. The definitions and descriptive statistics of each variable are shown in Table 2.

**Table 2.** Descriptive statistics.

| Variables | Obs | Mean | SD | Min | Median | Max |
|---|---|---|---|---|---|---|
| CLand | 1548 | 730.042 | 555.427 | 1.243 | 664.672 | 3784.382 |
| Treat·T | 1551 | 0.144 | 0.351 | 0.000 | 0.000 | 1.000 |
| EL | 1404 | 13.711 | 1.117 | 10.619 | 13.835 | 16.360 |
| PRCP | 1512 | 8477.941 | 1736.471 | 4203.795 | 8316.350 | 16,520.430 |
| PD | 1283 | 0.025 | 0.024 | 0.001 | 0.016 | 0.143 |
| FEL | 1398 | 12.371 | 0.567 | 10.486 | 12.380 | 13.996 |
| FRL | 1404 | 10.620 | 1.158 | 7.209 | 10.626 | 13.598 |
| AVPI | 1334 | 11.811 | 0.909 | 8.978 | 11.974 | 13.840 |

## 4. Results and Discussion

### 4.1. DID Regression Results

The regression results of the DID model are reported in Table 3. Columns (1) to (4) display the results without controlling for time-fixed and area-fixed effects, controlling for time-fixed effects only, controlling for area-fixed effects only, and controlling for both time-fixed and area-fixed effects, respectively. It can be seen that whether controlling for time-fixed or area-fixed effects, the poverty eradication policy significantly reduces the arable land area in poverty-stricken counties in the Qinba Mountains. Moreover, the regression results in column (4) indicate that the implementation of the poverty eradication policy successfully reduces 116.6 km$^2$ of arable land area in the contiguous impoverished areas in the Qinba Mountains.

**Table 3.** Main regression.

| | Arable Land Area | | | |
|---|---|---|---|---|
| | (1) | (2) | (3) | (4) |
| Treat·T | −174.9 *** | −123.4 *** | −152.5 *** | −116.6 *** |
| | (36.39) | (36.86) | (33.00) | (33.43) |
| EL | −189.7 *** | −109.0 *** | −4.840 | 32.41 |
| | (31.45) | (31.56) | (43.52) | (43.37) |
| PRCP | −0.0292 *** | −0.0377 *** | 0.0101 | 0.0115 |
| | (0.00722) | (0.00718) | (0.00981) | (0.0102) |
| PD | 1547.9 ** | −74.30 | −1245.6 * | −2953.2 *** |
| | (775.4) | (766.4) | (695.7) | (673.4) |
| FEL | 257.3 *** | 516.6 *** | 101.3 *** | 413.4 *** |
| | (35.13) | (43.20) | (39.21) | (47.82) |

**Table 3.** *Cont.*

| | Arable Land Area | | | |
| | (1) | (2) | (3) | (4) |
|---|---|---|---|---|
| FRL | −11.25 | −75.32 *** | −74.61 ** | −115.0 *** |
| | (22.02) | (23.05) | (29.96) | (30.73) |
| AVPI | 507.1 *** | 410.5 *** | 518.4 *** | 419.9 *** |
| | (26.97) | (25.85) | (25.25) | (24.73) |
| Time-Fixed Effects | No | Yes | No | Yes |
| Area-Fixed Effects | No | No | Yes | Yes |
| cons | −5455.6 *** | −7541.3 *** | −5686.2 *** | −8134.8 *** |
| | (409.0) | (484.2) | (403.0) | (494.1) |
| N | 1271 | 1271 | 1271 | 1271 |

Note: ***, ** and * indicate 1%, 5% and 10% significance levels, respectively.

## 4.2. Robustness Tests

### 4.2.1. Parallel Trend Test

To ensure that the regression results are stable and reliable, this study conducts a parallel trend test. The parallel trend can be used to test whether the experimental and control groups have the same trend before the exogenous shock occurs, that is, before the implementation of the poverty eradication policy. If they have the same trend between them, it means that the DID model obtained through the results is valid and can measure the net effect of the policy well [77]. Otherwise, the regression results are no longer representative of the policy effect, indicating that there are other factors that influence the change in arable land area.

From the test results in Figure 2, the difference between the experimental group and the control group is not significantly different from 0 before the time point of the policy, indicating that they are not different beforehand, so the model results are reliable. The difference between the experimental group and the control group becomes significant only after Post 2 because the policy has a lag, so its effect does not appear until two years after the policy is implemented.

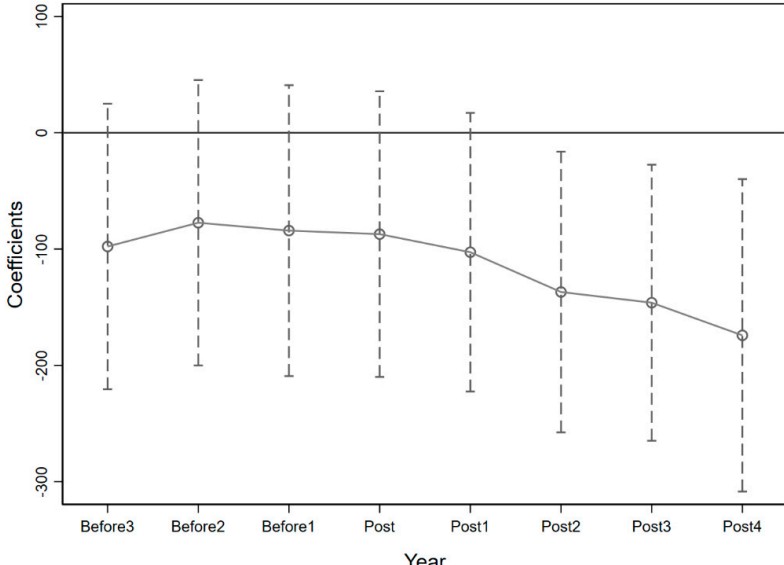

**Figure 2.** Parallel Trend Test.

### 4.2.2. Lagged Explanatory Variables

To address the endogeneity problem of the model, the core explanatory variables are with a one-period lag [78], which is commonly used in economics management studies and is effective in addressing reverse causality and endogeneity. The results are shown in column (1) of Table 4, where the cross-product term is still significant, which proves the robustness of the regression results.

**Table 4.** Robustness tests.

| | Lagged Explanatory Variables | Lagged Control Variables | Winsorize |
|---|---|---|---|
| L.Treat·T | −131.5 *** (36.64) | | |
| Treat·T | | −115.3 *** (33.23) | −92.70 *** (28.58) |
| EL | 20.80 (45.48) | | −11.61 (34.61) |
| PRCP | 0.0186 * (0.0111) | | 0.0120 (0.0082) |
| PD | −2927.1 *** (704.0) | | −2356.08 *** (635.40) |
| FEL | 431.8 *** (50.07) | | 369.01 *** (40.32) |
| FRL | −105.2 *** (31.60) | | −83.04 *** (24.73) |
| AVPI | 413.5 *** (26.56) | | 440.79 *** (20.57) |
| L.EL | | 32.80 (44.99) | |
| L.PRCP | | 0.0128 (0.0107) | |
| L.PD | | −2947.2 *** (723.1) | |
| L.FEL | | 422.6 *** (52.76) | |
| L.FRL | | −124.4 *** (33.28) | |
| L.AVPI | | 418.1 *** (22.64) | |
| Regional fixed effects | Yes | Yes | Yes |
| Time fixed effects | Yes | Yes | Yes |
| _cons | −8391.3 *** (526.8) | −8150.1 *** (535.9) | −7617.5 *** (351.9) |
| $N$ | 1129 | 1130 | 1271 |
| $R^2$ | 0.6111 | 0.6135 | 0.6627 |
| adj. $R^2$ | 0.6044 | 0.6069 | 0.6573 |

Note: *** and * indicate 1% and 10% significance levels, respectively.

### 4.2.3. Lagged Control Variables

In addition to the core explanatory variables, the control variables may also have endogeneity issues that can affect the regression results [79], so the regression results after

lagging all the control variables by one period are shown in column (2) of Table 4, where the effect of the poverty eradication policy on arable land area is still significantly negative, indicating the robustness of the regression.

### 4.2.4. Winsorize

To exclude the effect of extreme values, the variables are winsorize [80]. The values of variables less than 1% are replaced with values at 1%, and values greater than 99% are replaced with values at 99%. The regression results are presented in column (3) of Table 4 with significant cross-product terms, demonstrating the robustness of the model.

### *4.3. Mechanism Tests*
### 4.3.1. Intermediary Effect

After the above regressions and robustness tests, it is determined that the poverty eradication policy significantly negatively impacts the arable land area in the contiguous impoverished areas in the Qinba Mountains. In this section, the mechanism of this effect is further explored, i.e., the path through which the poverty eradication policy affects the arable land area in the Qinba Mountains.

As discussed above, the poverty eradication policy can potentially reduce the arable land area in the contiguous impoverished areas in the Qinba Mountains by increasing unit grain yield and retiring fragmented arable land. To further verify the mediating role of unit grain yield, a two-stage mediating effect model for validation is used [81–83].

Firstly, control the baseline regression to keep the same sample size as the regression after adding the mediating variables and then conduct the test for mediating effects. The first stage verifies whether the effect of the poverty eradication policy on unit food is significant. If significant, it is proceeded to the second stage. Otherwise, it is stopped.

$$UGY_{it} = \beta_0 + \beta_1 \text{Treat} \cdot \text{T} + \beta_2 \text{control}_{it} + \eta_t + \mu_i + \varepsilon_{it} \tag{2}$$

Based on the first stage's model, the second stage's model is improved by including arable land area as the explanatory variable and the cross-product term with unit grain yield as the explanatory variable. If the coefficient of unit grain yield is significant, it indicates a mediating effect. Otherwise, the mediating effect does not exist.

$$CLand_{it} = \beta_0 + \beta_1 \text{Treat} \cdot \text{T} + \beta_2 UGY_{it} + \beta_3 \text{control}_{it} + \eta_t + \mu_i + \varepsilon_{it} \tag{3}$$

The results of the intermediate effect test are displayed in columns (1)–(3) of Table 5. Column (2) illustrates that the poverty eradication policy can significantly increase the unit grain yield in the Qinba Mountain contiguous poverty area. In column (3), both the coefficients of the cross-product term and the unit grain yield are significantly negative. It indicates that the unit grain output can significantly reduce the cultivated land area, and the poverty alleviation policy reduces the cultivated land area in the Qinba Mountain Area by increasing the unit grain output.

**Table 5.** Mechanism Test.

|  | (1) Cland | (2) UGY | (3) Cland | (4) RSEI |
|---|---|---|---|---|
| Treat·T | −123.3 *** (33.95) | 35.99 *** (8.772) | −88.65 *** (32.96) |  |
| UGY |  |  | −0.912 *** (0.0849) |  |
| c_Treat·T |  |  |  | 107.3 *** (40.54) |

**Table 5.** *Cont.*

|  | (1) Cland | (2) UGY | (3) Cland | (4) RSEI |
|---|---|---|---|---|
| c_RSEI |  |  |  | −1756.9 *** |
|  |  |  |  | (154.2) |
| Interact |  |  |  | −1226.6 *** |
|  |  |  |  | (275.2) |
| EL | 26.86 | −26.73 *** | 1.154 | −59.81 * |
|  | (44.09) | (9.941) | (42.34) | (35.82) |
| PRCP | 0.0134 | 0.00734 *** | 0.0205 * | 0.0462 *** |
|  | (0.0107) | (0.00252) | (0.0111) | (0.00907) |
| PD | −2696.6 *** | 4674.6 *** | 1800.7 ** | −6902.1 *** |
|  | (684.6) | (284.2) | (787.4) | (713.2) |
| FEL | 430.6 *** | −71.29 *** | 362.0 *** | 311.6 *** |
|  | (51.01) | (11.69) | (48.72) | (38.48) |
| FRL | −126.4 *** | −6.725 | −132.9 *** | −22.85 |
|  | (32.68) | (7.282) | (32.51) | (23.92) |
| AVPI | 431.8 *** | 36.31 *** | 466.7 *** | 412.7 *** |
|  | (25.63) | (6.730) | (25.57) | (23.41) |
| Time Fixed Effects |  | Yes | Yes | Yes |
| Regional Fixed Effects |  | Yes | Yes | Yes |
| cons | −8304.9 *** | 871.7 *** | −7466.3 *** | −6764.8 *** |
|  | (530.9) | (109.4) | (502.4) | (332.4) |
| N | 1212 | 1212 | 1212 | 1271 |

Note: ***, ** and * indicate 1%, 5% and 10% significance levels, respectively.

### 4.3.2. Moderating Effect

In this section, we will analyze what factors can change the magnitude of the effect of poverty eradication policies on the arable land area in the Qinba Mountains. As discussed in Theoretical Mechanisms, in China, a large population combined with limited control by local governments has occupied a large amount of arable land, leading to the deagriculturalization of arable land, non-food, and abandonment of arable land. After the drastic degradation of arable land quality and the inability of the existing arable land output to meet the livelihood of the poor, the people in poverty-stricken areas will further deforest and clear new arable land to earn a living. These phenomena are undoubtedly a vicious circle of continuous environmental destruction and sacrifice of ecological resources for poverty-stricken areas, which will cause huge externalities to biodiversity and climate change and is extremely unfavorable to sustainable development.

Poverty alleviation will take ecological poverty alleviation as a principle and insist on poverty alleviation without harming the environment. Therefore, the government of poverty-stricken counties will promote local land preparation work to reduce ecological risks [84] and further promote the reduction of arable land area by returning farmland to forest and grass [85]. This study regulates the ecological environment quality through the regulation effect model.

The moderating effect model is based on the baseline regression model by introducing the moderating variables and the cross-product term composed of the moderating variables and the core explanatory variables [86]. What should be paid attention to is the coefficient of this cross-product term $\beta_3$, the coefficient of the core explanatory variables will become inaccurate at this point. If the cross-multiplication term is the same as the coefficient of the explanatory variables in the main regression, it indicates that the moderating variables can enhance the main regression effect; otherwise, the moderating variables will weaken the main regression effect [87]. At the same time, to avoid multicollinearity of the core explanatory variables, the moderating variables, and the cross-product term composed of the core explanatory variables and the moderating variables, the core explanatory variables

and the moderating variables are centralized, which would not have any effect on the test results of the moderating effect [88].

$$CLand_{it} = \beta_0 + \beta_1 c\_Treat \cdot T + \beta_2 c\_RSEI_{it} + \beta_3 Interact_{it} + \beta_4 control_{it} + \eta_t + \mu_i + \varepsilon_{it} \quad (4)$$

We choose the remote sensing ecological index as a proxy variable for ecological quality. The remote sensing ecological index consists of four dimensions, humidity, greenness, temperature and dryness, which can effectively measure ecological quality and is highly recognized by scholars and widely used in environmental economics research [23,67].

Column (4) of Table 5 displays the regression results of the moderating effect model. The coefficient of the cross-product term is significantly negative, the same as the coefficient of the baseline regression. It indicates that based on the pressure to improve the quality of the ecological environment. Poverty-stricken county governments will further strengthen the return of cultivated land to forest and grass and ecological restoration of high investment and low yield as well as saline land, thus promoting the reduction of cultivated land area.

### *4.4. Heterogeneity Analysis*

Since regional environmental heterogeneity and resource endowment, local economic development, etc., can affect the policy implementation effect, it is necessary to conduct heterogeneity analysis for the baseline regression results. This study is analyzed from four perspectives, including regional elevation, unit agricultural machinery power, rural per capita income, vegetation cover, and upgrading industrial structure.

### 4.4.1. Heterogeneity Test Based on Regional Elevation

Regional elevation determines the distribution of arable land area [89,90]. Usually, the spatial distribution of settlements within the agricultural land system in plain areas is uniform, and the intensification of arable land is significant, which is easier to integrate than in mountainous areas. Therefore, the change of arable land area in plain areas will be more significant, and the intensification of arable land resources will mainly occur in areas with lower elevations. As shown in column (1) of Table 6, the arable land area in poverty-stricken counties at lower elevations is significantly lower than in higher elevations.

**Table 6.** Heterogeneity analysis.

| | (1) Elevation | | (2) UAMP | | (3) RPCI | | (4) NDVI | |
|---|---|---|---|---|---|---|---|---|
| | Low | High | Low | High | Low | High | Low | High |
| Treat·T | −236.7 *** | 52.73 * | −5.453 | −215.8 *** | −71.84 ** | −111.6 * | −114.9 ** | −6.465 |
| | (42.92) | (29.46) | (42.09) | (49.88) | (28.76) | (59.04) | (48.40) | (39.75) |
| EL | −49.43 | −7.876 | 172.0 *** | −41.37 | -26.49 | -36.36 | −320.3 *** | 171.2 ** |
| | (55.33) | (48.15) | (65.33) | (59.14) | (55.87) | (56.56) | (47.39) | (70.20) |
| PRCP | −0.00391 | 0.00802 | 0.00178 | 0.0244 * | 0.0324 *** | 0.00165 | -0.000949 | 0.0170 |
| | (0.0123) | (0.00889) | (0.0141) | (0.0140) | (0.00946) | (0.0163) | (0.00875) | (0.0171) |
| PD | −4029.1 *** | 30,050.5 *** | 459.3 | −2851.7 *** | 6116.1 *** | −3838.3 *** | −4610.6 *** | 37,761.9 *** |
| | (718.5) | (3136.1) | (1128.8) | (1007.9) | (1061.3) | (832.6) | (543.5) | (2543.9) |
| FEL | 747.1 *** | 16.28 | 395.8 *** | 450.3 *** | 178.7 *** | 604.5 *** | 603.9 *** | 92.08 |
| | (70.98) | (57.92) | (67.72) | (69.39) | (52.88) | (80.84) | (61.48) | (66.13) |
| FRL | −283.3 *** | 138.1 *** | −118.1 *** | −137.4 *** | 54.99 * | −163.1 *** | 1.423 | −114.6 *** |
| | (44.45) | (29.77) | (41.37) | (44.07) | (30.05) | (43.58) | (33.26) | (43.35) |
| AVPI | 486.2 *** | 148.2 *** | 305.1 *** | 417.2 *** | 360.4 *** | 432.7 *** | 478.5 *** | 177.2 *** |
| | (26.23) | (31.76) | (44.46) | (27.42) | (47.76) | (27.43) | (22.21) | (43.36) |
| Regional Fixed effects | Yes | Yes | Yes | Yes | Yes | Yes | Yes | Yes |
| Time Fixed Effects | Yes | Yes | Yes | Yes | Yes | Yes | Yes | Yes |
| _cons | −9770.9 *** | −2880.1 *** | −8393.3 *** | −7388.2 *** | −5987.1 *** | −8989.1 *** | −7168.6 *** | −4179.4 *** |
| | (741.5) | (551.8) | (754.4) | (656.2) | (474.3) | (729.2) | (433.6) | (801.9) |
| *N* | 875 | 396 | 747 | 524 | 587 | 684 | 585 | 686 |

Note: ***, ** and * indicate 1%, 5% and 10% significance levels, respectively.

### 4.4.2. Heterogeneity Test Based on Unit Agricultural Machinery Power

Column (2) of Table 6 presents the regression results of heterogeneity of unit agricultural machinery power (UAMP). The unit of agricultural machinery power represents the regional agricultural mechanization level [91]. The higher the agricultural mechanization power, the higher the local agricultural development level. What's more, with higher agricultural mechanization power, the efficiency of arable land use will increase significantly, and governments in poverty-stricken areas will be able to reduce the use of arable land while ensuring the same yield [92]. Therefore, in areas with higher levels of mechanization, the arable land area is declining more significantly.

### 4.4.3. Heterogeneity Test Based on Rural Per Capita Income

Column (3) of Table 6 illustrates that in areas with higher rural per capita income (RPCI) levels, the arable land area reduces more rapidly. The rise in income level represents an improvement in the economic situation of the poverty-stricken [93], and the poverty eradication policy has brought more non-farm employment opportunities to the people in the Qinba Mountains and increased the local per capita income level. When income levels are low, people tend to keep land for self-sufficiency to resist uncertainty for precaution and risk [94]. So, when economic conditions improve, and poor people face more employment options, fewer people will keep their land.

### 4.4.4. Heterogeneity Test Based on NDVI

The normalized difference vegetation index (NDVI) represents the ecological level of the area to some extent [95,96]. In column (4) of Table 6, the lower the NDVI, the more the reduction of arable land area. For poverty-stricken areas, returning farmland to forest and grass is an important method to achieve ecological poverty alleviation. Low NDVI represents that the local green area is less, and the ecological environment needs to be improved [97,98]. The government will be more proactive in promoting ecological restoration, such as returning farmland to the forest.

## 5. Conclusions and Policy Recommendations

### 5.1. Conclusions

From the perspective of policy evaluation of the impact of poverty eradication policy on arable land in poverty-stricken areas, this paper adopts the DID model to study the impact of the policy on the amount of contiguous impoverished areas in the Qinba Mountains by a quasi-natural experiment of poverty eradication policy implementation, verifying the effects of the policy through robustness tests and controlling for potential endogeneity variables. As an international case of a more successful public policy focused on poverty eradication, this study summarizes the successful practice in China can provide a useful reference for poverty eradication actions in other countries and regions and lays a favorable foundation for achieving sustainable development.

This study reveals that policy implementation has led to a reduction in the cultivated land area in impoverished regions, and this reduction is attributed to improvements in yield per unit and environmental quality. Firstly, policies have raised yield per unit through the construction of high-standard farmland and the promotion of efficient agricultural technologies. This transformation has alleviated the reliance of impoverished populations in these areas on excessive new land reclamation to sustain their livelihoods. Secondly, policies encourage ecological restoration of degraded and saline-alkali soils, enhancing the environment while reducing cultivated land area. These dual measures have collectively contributed to the reduction in cultivated land area in impoverished regions without compromising production and the livelihood needs of the local population.

The findings of this study shed light on the pathways through which land management can simultaneously address the livelihood needs of impoverished populations and protect the ecological environment. It underscores the importance of well-designed policies safeguarding food security in impoverished areas while efficiently utilizing land

resources for ecological conservation. In future policy formulation, it is crucial to consider the coordination of measures aimed at increasing crop yields and implementing land retirement for afforestation and grassland restoration, thereby maintaining a balance between livelihoods and ecological preservation. Additionally, it is imperative to strengthen the monitoring of land use dynamics in impoverished regions and conduct timely assessments of policy effectiveness.

### 5.2. Contributions and Limitations

### 5.2.1. Contributions

Although numerous studies have focused on China's poverty alleviation policies, limited attention has been given to the impact of these policies on the reduction of cultivated land in impoverished areas. This paper addresses this gap by concentrating on the relationship between changes in cultivated land area and poverty alleviation policies. Secondly, this study contributes by elucidating the mechanisms through which poverty alleviation policies lead to a reduction in cultivated land. It clarifies how the simultaneous attainment of livelihood needs for impoverished populations and the protection of the ecological environment can be achieved through poverty eradication. This enrichment of the content and scope of poverty governance offers empirical evidence to policymakers, providing reference and insights for the sustainable development of impoverished regions. It facilitates the advancement of strategies for sustainable development.

### 5.2.2. Limitations

Despite the examination of the impact of China's poverty alleviation policies on the reduction of cultivated land in impoverished areas, there are still aspects that have not been comprehensively investigated. For instance, policies such as social security, industry-targeted poverty alleviation, and employment-focused poverty alleviation may affect the study outcomes. In future research, it is imperative to consider these limitations and conduct in-depth investigations into the impact of other poverty alleviation strategies on the reduction of cultivated land in impoverished regions. By comprehensively assessing and comparing the effects of different factors, a more comprehensive evaluation of the efficacy of various poverty alleviation strategies can be conducted, thus providing valuable insights for the formulation of more effective poverty alleviation policies.

### 5.3. Policy Recommendations

The governance and conservation of cropland in poverty-stricken areas has been a governance problem in many countries and regions worldwide. Most of the cropland in poverty-stricken areas worldwide faces problems such as the vicious cycle of low production with high input-low output, land degradation and land salinization. It will indirectly encourage the population in poverty-stricken areas to destroy grasslands and woodlands and convert them into arable land for extensive agricultural production to earn a living, which is an extremely harmful and destructive act for the ecological environment. Therefore, in the future, it is necessary to improve the infrastructure in poverty-stricken areas, build high-quality farmland, and improve the soil to increase the unit yield of arable land to fundamentally stop the people in poverty-stricken areas from expanding cultivation for livelihood. Moreover, it is of great significance to increase the monitoring of arable land in poverty-stricken areas, and actively guide farmers to implement crop rotation and fallow initiatives to continuously restore the quality of arable land and provide opportunities for sustainable cultivation.

**Author Contributions:** All authors contributed to the study's conception and design. Conceptualization, writing—review and editing, R.R. and J.X.; methodology and formal analysis, data curation and writing—original draft preparation, L.H.; investigation, project administration, T.L. and Y.C. All authors have read and agreed to the published version of the manuscript.

**Funding:** This research was funded by the National Natural Science Foundation of China (Grant No. 72074035), Graduate Research and Innovation Foundation of Chongqing, China (CYB22057), and Fundamental Research Funds for the Central Universities (2020CDJSK01PT20, 2022CDSKXYGG006).

**Data Availability Statement:** The data that support the findings of this study are available from the corresponding author upon reasonable request.

**Conflicts of Interest:** The authors declare no conflict of interest.

## Notes

[1]  Poverty-stricken counties referred to in the article are those counties that were designated as national poverty-stricken counties by the Chinese government in 2015 and have achieved poverty eradication under the promotion of poverty eradication policies.

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
