# Peer review of "Why Have China’s Poverty Eradication Policy Resulted in the Decline of Arable Land in Poverty-Stricken Areas?"

_land, doi:10.3390/land12101856_

Round 1
Reviewer 1 Report
1) The question "Why have China's poverty eradication policy resulted in the decline of arable land in the poverty-stricken areas?", is not clearly enough answered in the Conclusions.
2) To add some maps would help the reader understand better the location and geography of studied region.
3) Only the aspects of the poverty eradication policy directly related to the arable land are mentioned; are there no other strategies for the eradication of poverty in the mentioned policy? Economic activities that could substitute or complement agriculture?
4) Are there some special environmental values in the region demanding ecological restauration that the mentioned policy could support?
5) Food provision and food security are mentioned; how are they guaranteed for the population through the policy?
6) The theory framework is quite "weak", especially regarding socio-cultural and nature-culture factors; how to tackle these questions in the implementation of politics?
There are parts of the text with tautology that should be corrected.
Author Response
Dear reviewer, thank you for your valuable comments on our article, we have responded to your comments and revised one by one, the specific content has been uploaded to the system in the form of attachments, thank you again for your help!

Reviewer 2 Report
Dear Authors,
I have read your manuscript on a vital topic to gauge the effectiveness of the policy instrument. However, I feel the manuscript lacks in the following areas and requires a straightforward thought process before publishing.
The areas where the manuscript requires a significant amount of revision are the following;
1. You have clearly stated the hypothesis and objectives of the study which is good but you do not mention whether the previous literature has any conclusion on this topic.
2. Second, I feel too much difficulty understanding the methods section and explanation of the variables. Please revise it thoroughly and make the section more simple, easily readable, and understandable.
3. The same observation I have on the results section Please revise it and state clearly what are your key results and what are the policy implications for the future. In its current form, this section is too much descriptive in nature and lacks clarity and consequently does not flesh out the key results clearly for the reader. Always keep in mind your reader is always not a subject matter specialist.
Some of my minor observations on the manuscript are;
Ln 47 What do you mean by various natural environments? It is not clear?
Ln 49 Rather than the previous process. I think previous phases is a better expression
Ln 67-70 Unfortunately, all these examples are unable to unfold what actual methods were employed for strengthening and protecting arable land quality
Table 1. Variables units are worth mentioning to understand these descriptive statistics. Without units, this table doesn't make any sense.
Ln 224 The "Double Difference Model" is the same as the "Difference in Difference" Model?
The Manuscript needs a thorough revision for the clarity of expression and argument. In some places, more direct sentence choices can solve this issue.
Author Response

(The authors gave the same response as above.)

Reviewer 3 Report
This is an interesting and meaningful study. This article examines the impact of policy implementation on arable land by Difference-in-Differences model, and uses the mediating and moderating models to test the policy’s mechanism on arable land. In general, the research ideas are clear, the methods are appropriate. It is suggested to minor revision. Some suggestions are for reference:
1. In the second paragraph of the introduction, it is suggested to initially discuss changes in arable land from the perspective of impoverished regions before introducing the specific case of the Qinba Mountain area. This would create a smoother transition and avoid the abrupt introduction of the Qinba Mountain area at the outset, which currently appears disjointed.
2. The theoretical analysis section suggests further strengthening. The first is a more detailed introduction of the logic and mechanism between poverty eradication policy and the decline of arable land. The second is the theoretical mechanism that increasing the unit grain yield and improving the ecological environment will reduce the arable land area.
3. The discussion section needs to be strengthened. For example, how is this study different from other studies?
4. In the section discussing control variables, the use of local GDP as a control variable is mentioned, but there is a lack of explanation regarding why GDP would influence changes in arable land area. Please include relevant explanations.
5. Please thoroughly proofread the language and writing style once more to enhance the article's readability. For example there is a spelling error in the line 143.
Author Response

(The authors gave the same response as above.)

Round 2
Reviewer 1 Report
* Please, there are still some tautology in the introductory part of the text. It should be attended.
* I consider that the presentation of the study area should be located differenty in the text; it appears somehow abruptly and without connection to the preceding paragraph.
* Please, there are still some tautology in the introductory part of the text. It should be attended.
Author Response
Dear reviewer,
Thank you very much for your valuable comments, we have revised the paper according to your comments, and the specific responses and detailed changes have been uploaded in the form of attachments. Thank you again for the help you have given us!

Reviewer 2 Report
Dear Authors,
Thank you for considering my comments valuable and revised accordingly.
Author Response
Dear Reviewer:
Thank you very much for recognizing us and thank you again for all the help you have given us!